# Towards the Future of Polymeric Hybrids of Two-Dimensional Black Phosphorus or Phosphorene: From Energy to Biological Applications

**DOI:** 10.3390/polym15040947

**Published:** 2023-02-14

**Authors:** Avneesh Kumar, Dong Wook Chang

**Affiliations:** Department of Industrial Chemistry and CECS Research Institute, Pukyong National University, Busan 48513, Republic of Korea

**Keywords:** two-dimensional materials, black phosphorus, phosphorene, functional polymers, nanocomposites, battery materials, cancer treatment, drug delivery, water purification

## Abstract

With the advent of a new 2D nanomaterial, namely, black phosphorus (BP) or phosphorene, the scientific community is now dedicated to focusing on and exploring this 2D material offering elusive properties such as a higher carrier mobility, biocompatibility, thickness-dependent band gap, and optoelectronic characteristics that can be harnessed for multiple applications, e.g., nanofillers, energy storage devices, field effect transistors, in water disinfection, and in biomedical sciences. The hexagonal ring of phosphorus atoms in phosphorene is twisted slightly, unlike how it is in graphene. Its unique characteristics, such as a high carrier mobility, anisotropic nature, and biocompatibility, have attracted much attention and generated further scientific curiosity. However, despite these interesting features, the phosphorene or BP poses challenges and causes frustrations when it comes to its stability under ambient conditions and processability, and thus in order to overcome these hurdles, it must be conjugated or linked with the suitable and functional organic counter macromolecule in such a way that its properties are not compromised while providing a protection from air/water that can otherwise degrade it to oxides and acid. The resulting composites/hybrid system of phosphorene and a macromolecule, e.g., a polymer, can outperform and be exploited for the aforementioned applications. These assemblies of a polymer and phosphorene have the potential for shifting the paradigm from exhaustively used graphene to new commercialized products offering multiple applications.

## 1. Introduction

In the family of low dimensional materials, particularly inorganic ones, two-dimensional (2D) black phosphorus or BP, an allotrope of phosphorus, has gained tremendous attention due to its elusive properties [1,2,3,4,5,6,7,8,9,10,11,12]. The few-layered form of BP is commonly known as phosphorene [13,14]. Being a competitor of graphene, it has distinct features such as a thickness-dependent band-gap (0.3 to 2 eV), high carrier mobility (up to ~1000 cm^2^/V·s) at room temperature, anisotropic nature (mechanical, thermal, optical, and electrical), and biocompatibility. The applications of BP nanosheets are mainly being explored in photoelectronic devices, biomedicine, catalysis, and energy storage [15,16,17,18,19,20]. In 2D BP nanosheets or phosphorene, the phosphorus atoms are hexagonally packed, leading to the formation of a puckered ring, as observed in its crystal structure. Initially, BP was prepared from its precursor white phosphorus under a high pressure and temperature (1.2 GPa and 200 °C, respectively) [21,22,23]. Since then, several other methods have emerged to obtain the bulk BP as well its nanosheets with a certain thickness [24,25,26,27,28,29,30]. Currently, the nanosheets of BP can be afforded via mechanical and solvent mediated exfoliation [24,25,26,27,28,29,30]. However, there are still some challenges to synthesize the BP nanosheets with an improved quality and in a larger quantity for commercial applications. Nevertheless, the overall properties of phosphorene can be optimized accordingly by reducing its thickness during the synthesis. One of the greatest challenges that the researchers working with phosphorene are facing is its instability, due to which the industrial usage of BP nanosheets remains still a milestone to be achieved [31,32,33,34,35,36,37,38]. When exposed to the air or ambient conditions, the BP nanosheets can degrade by reacting with oxygen and produce the oxide of phosphorus, thereby lowering the performance of a fabricated device [34,35,36,37,38]. In order to protect the BP nanosheets from the oxidative degradation and to maintain its tailored properties, the efforts to couple or sheath it with the organic materials, particularly polymers, have been ongoing for a considerable amount of time [39,40,41,42].

Polymers or macromolecules with the active functionality and controllable characteristics have become an essential commodity in our daily life [43,44,45,46,47,48,49,50,51,52,53,54,55,56,57,58,59,60,61,62,63]. These large molecules with guided attributes and a complementary chemical component can provide phosphorene nanosheets with a protective environment and also contribute further towards the performance of BP nanosheets by increasing the photocatalytic activity, in case of a water split process, or biocompatibility for drug delivery or sensing applications [64,65,66,67,68,69,70,71]. Further research with regard to 2D BP nanosheets and suitable polymers for a commercial purpose is still necessary so that phosphorene-based devices can be fabricated and used on a large scale in energy production and biomedical applications. Therefore, with a focus on related research and development efforts, this review mainly aims at the hybrid materials systems composed of BP nanosheets and active polymers in order to discuss the current on-going attempts to exploit BP nanosheets, in combination with polymers, for multiple applications. Figure 1 depicts the BP nanosheets, quantum dots, their hybrids with the counterpart polymers, together with the current applications of such hybrid material systems [72,73]. Another emphasis of this review is directed on the synergetic function of various polymers, such as conducting or being peptide-based in combination with the BP nanosheets or phosphorene (Figure 1). This review shows further perspectives for developing the hybrid system of BP nanosheets and polymers, offering a high performance in a device. An overview of various polymers with the active functional groups attached or adsorbed to the surface of BP nanosheets or quantum dots is also shown in Figure 1 for a better understanding.

## 2. BP Nanosheets or Phosphorene Synthesis and Characterization

As mentioned earlier, bulk black phosphorus (BP) can be prepared from white phosphorus under a high pressure of about 1.2 GPa and a temperature of 200 °C [21,22,23]. The few-layered BP nanosheets that are the focus of interest and are usually synthesized from its bulk BP precursor. In 2014, the scotch tape-based mechanical exfoliation of bulk BP crystal (bottom-down synthesis) was carried out to obtain the flakes of BP [74]. These few-layer phosphorene or BP flakes were fabricated on a silicon wafer surface for FETs (field effect transistors). Usually, the exfoliation of bulk BP can be complex, owing to the stronger interlayer interactions (~151 meV per P atom) and electron lone pairs on phosphorus atoms. Thus, liquid phase exfoliation (bottom-up synthesis) is sought to delaminate the bulk BP, yielding the few-layer nanosheets in a reasonable quantity or large scale [75]. Various organic solvents, such as N-methyl-2-pyrrolidone (NMP), in [76,77,78], N-vinyl pyrrolidone (NVP), in [79], dimethyl formamide (DMF), in [80,81], dimethyl sulfoxide (DMSO), and in [82,83,84,85,86,87,88], isopropyl alcohol (IPA), have been used for preparing 2D BP nanosheets. Figure 1 shows the exfoliation of BP under sonication into different shapes and sizes. In a typical liquid phase process, the exfoliation of BP is conducted in the presence of a specific solvent under ultra-sonication, which can lead to the formation of 2–5 layered phosphorene depending on the solvent and time taken for ultra-sonication.

Table 1 summarizes the solvent exfoliation methods for preparing BP nanosheets in different solvents and dispersants. The nature of the solvent and frequency of sonication cn directly impact the size of the BP nanosheets, which could be very crucial for their certain applications. In a separate method, both NaOH and NMP were used for exfoliating the BP to produce the phosphorene with an improved dispersion and stability in water [89,90,91]. It was observed that the high power of ultra-sonication can greatly influence the exfoliation of BP by destroying the weak van der Waals interaction between the sheets of bulk BP, thus allowing the evolution of thin layers of BP. Apart from the nanosheets, the BP can be transformed into quantum dots (QDs) by following a similar liquid phase exfoliation procedure and optimizing the sonication power and time [92,93]. These BPQDs could be as small as 2.6 nm with a thickness of 1.5 nm. To improve the scale and stability of BP nanosheets via exfoliation, third-phase dispersants such as surfactants, polymers, and ionic liquids have also been employed and reported already in the literature [94,95]. Following the bottom-up synthesis, a facile, scalable, and low-cost approach for producing BP nanosheets in a large quantity has been described. In this approach, the one-pot wet-chemical synthesis of BP nanosheets in gram-scale quantities was conducted by using white phosphorus and ethylenediamine as a solvent [96]. Based on the results, it was concluded that BP nanosheets can be manufactured via a facile solvothermal process using white phosphorus at a low temperature. For the morphology of 2D BP nanosheets, in Raman spectra, three distinct peaks at 360.2, 437.5, and 464.6 cm^−1^ (Figure 2) are seen and ascribed to the characteristic A_g_^1^, B_2g_, and A_g_^2^ vibrational modes of BP. In some cases, when BP nanosheets are partially oxidized, a peak at about 400 cm^−1^ could be observed for the P–O species.

The surface morphology and crystal structure of 2D BP nanosheets are usually investigated under AFM and TEM. Under transmission electron microscopy, a two-dimensional structure of the nanosheets can be observed, in which the lattice and its interlayer spacing can be estimated (Figure 2). As explained, the interlayer or *d* spacing of about 0.507 nm corresponds to the (020) plane of the orthorhombic morphology of BP (Figure 3). The thickness and complementary structure of BP nanosheets can be also studied by AFM, which can further provide information regarding the thickness and number of layers in BP nanosheets. Usually, for the thickness of 1–15 nm, about 2–28 atomic layers of BP are stacked together, forming the two-dimensional structure.

The thermogravimetric analysis of BP nanosheets or phosphorene has suggested their stability up to 350 °C [97]. At ambient conditions, when exposed to the atmosphere or water, BP nanosheets can form oxides (P_x_O_y_) and phosphoric acid. Such modifications can lead to the perturbation in molecular orbitals and thus alter the electronic properties. In X-ray photoelectron spectroscopy analysis, the appearance of two peaks at 129.5 and 130.4 eV corresponding to the single spin-orbit P 2_P_^3/2^ and doublet P 2_P_^1/2^ have indicated the crystalline nature of BP nanosheets (Figure 3a) [98,99,100]. However, the sub-bands for the oxidized species, such as P_2_O_4_ or P_2_O_5_, pointed out the presence of defects that may have originated during the exfoliation. Nevertheless, for some cases, the content of a P_x_O_y_ has been reported to be as high as ~15%, which can be reduced or diminished by stabilizing the BP nanosheets by solvent molecules or other dispersants.

**Figure 3 polymers-15-00947-f003:**
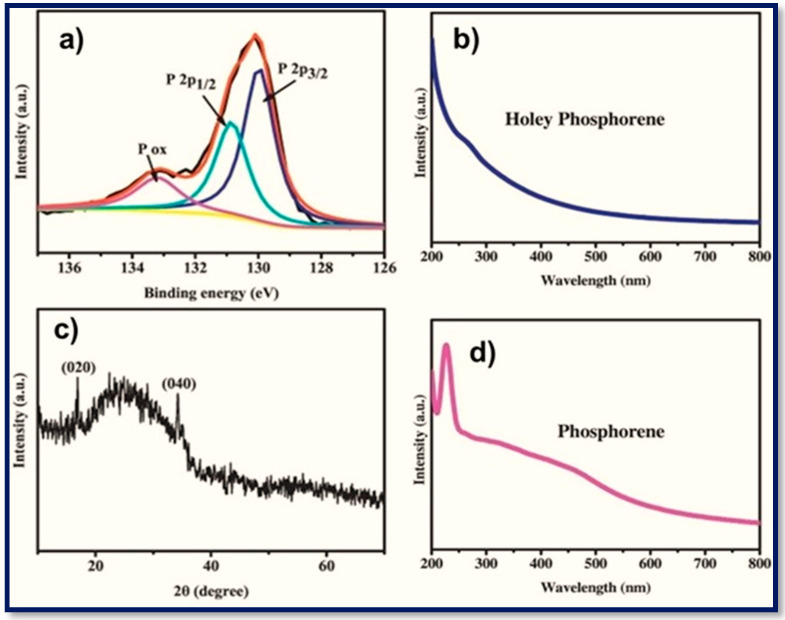
XPS, UV-VIS, PL, and X-ray diffractions of BP nanosheets. (**a**) High-resolution XPS spectra of P2p for the holey phosphorene; XRD pattern of the holey phosphorene, (**b**) The UV–Vis absorption spectra of holey phosphorene obtained by electrochemical assistance and phosphorene obtained by ultrasonic exfoliation, (**c**) X-ray diffractions of BP nanosheets, and (**d**) PL of BP nanosheets. Copyright 2018, Elsevier Inc. Reproduced with permission from Electrochemistry Communications [98], Copyright 2018, Elsevier Inc. Amsterdam, The Netherlands.

The X-ray experiments have shown that BP or phosphorene have arrays of phosphorus atoms hexagonally packed in its layer (Figure 3). However, the hexagonal ring of phosphorus atoms is slightly puckered. Discussing the electronic properties of phosphorene, as stated earlier, the band-gap for BP nanosheets is directly related with the thickness and can vary depending on the number of the layers. In Figure 3b–d, the optical properties of phosphorene are displayed. In the literature, the band gap of 0.3 to 2 eV is reported for the bulk and monolayer, respectively [101,102]. In UV–Vis spectrum of BP nanosheets, typically a broad absorption band in the visible and NIR region, demonstrates that BP nanosheets can harness solar energy and produce the sufficient amount of electric conductivity necessary for their applications in photocatalysis, field effect transistors, and biomedical applications such as phototherapy (Figure 3b–d).

As proposed earlier, phosphorene as a competitor of graphene, can provide better modulated electronic properties and suitability for integrating in a biological system. In order to support this view, a comparison between the properties of phosphorene and graphene is displayed in Table 2.

As phosphorene can react with O_2_ and H_2_O, it can, therefore, degrade in a biological system, whereas graphene with a higher content of the aromatic carbonaceous fraction does not degrade easily and thus is more toxic towards living cells. The optical absorption ability of phosphorene is also superior due to the absence of impurities; thus, it can be a promising material for an FET device compared to graphene.

## 3. Nanocomposites or Hybrids of Polymers and BP Nanosheets

For processing, stabilizing, and modulating the properties of 2D BP nanosheets, it is highly imperative to combine an appropriate polymer with them either by a covalent linkage or simply by the adsorption process. A schematic diagram to explain the functionalization of nanosheets with polymer chains via a covalent linkage or adsorption is shown in Figure 4. To overview the synthesis, a dispersion of as-obtained BP nanosheets from the exfoliation method discussed previously is taken to react the surface with an active initiator molecule. The surface of the nanosheets becomes decorated with the initiator molecules, through which a desired monomer is then polymerized in a suspension to afford the BP nanosheets–polymer.

Alternatively, both the polymer and BP nanosheets are mixed in a solvent, in which polymer chains with active side groups are adsorbed on the surface of nanosheets via non-covalent interactions, resulting in the composite of a BP nanosheet–polymer which can be used for processing in device fabrication (Figure 4). Such hybrids of BP nanosheets and polymers can be easily fabricated on a device for commercial purpose or real-world applications. Polymers with unique features and ’active’ functional groups are used for decorating or anchoring the 2D surface of BP nanosheets. In a number of reports, the chemically different polymer chains have also been linked to the surface of nanosheets via a covalent bond. In these examples, different types of polymers, for instance, conjugated polyacetylene, and polyglycerol chains were linked to the surface of BP nanosheets by following diazonium salt chemistry and ring-opening polymerization, respectively [103,104,105]. The further covalent functionalization of BP nanosheets with small organic molecules, such as porphyrins, has been also described in the literature. As the focus of this review is directed on the composites of polymers and BP, herein, several successful attempts using different polymers with ionic moieties or peptide building blocks in combination with BP nanosheets are described, including their applications. The following criteria can be applied for selecting a suitable polymer counterpart that can be coupled with the BP surface for achieving the maximum efficiency: (1) it must bear the appropriate functional group to interact and break down the interlayer interactions within the layered nanostructured BP, (2) the polymer must not degenerate the structure and properties of the individual components and must rather act synergistically with the BP nanosheets or nanodots, (3) depending on the application, the polymer must enhance the processability of the relevant composite for its fabrication in a device, and (4) for biological applications, the polymer must be biocompatible and be responsive to certain external stimuli or a signal.

### 3.1. Hybrids of Polymers and BP Nanosheets as Flame Retardants

The ecofriendly and inexpensive fire retardants, such as phosphorus-, nitrogen-, and silicon-containing materials, are sought as an alternative to the harmful and high-cost halogenated ones. In this regard, phosphorus nitrogen-based intumescent flame retardants (IFRs) are potential flame retardants for PU, benefiting from their halogen-free, low-toxicity, low-cost, and impressive fire hazard control [106]. The mechanism of flame retardancy undergoes a number of processes in which physical and chemical changes take place. For an overview, initially, it occurs with a combustion (endothermic) of the material when the temperature reaches its ignition point. During this step, the flame-retardant material is able to absorb a fraction of heat (endothermic reaction) released by the burning material, thereby reducing the surface temperature of the burning material. As a result, the fire and generation of smoke are prevented. Usually, the combustion proceeds with the formation of free radicals. Therefore, in order to decrease the flame density and burning rate of the combustible materials, the later must react with these free radicals, which in turn can extinguish the combustion process. Furthermore, a shielding layer is developed by the combustible material to minimize the exposure of the surface to the air so that the combustion gas does not escape, and fire can be stopped. Owing to the capability of black phosphorus to react with these free radicals, it is also the fire-retardant material of choice. It can promote the formation of char and capture the free radicals during the combustion. The layered structure of BP also acts as a physical barrier to oxygen and heat, allowing for the combustion process to slow down.

Recently, in an attempt to create the self-assemblies of BP nanosheets and a thermoplastic polymer, e.g., polyurethane (PU), the ionic liquid molecule 1-allyl-3-methylimidazolium chloride was employed to cover the surface of the nanosheets, which in the next steps were linked with PU chains (Figure 5) [107,108,109,110].

The vinyl group in ionic liquid was polymerized via a free radical process, resulting in the polymerized coating of ionic liquid on the 2D surface of nanosheets. These functionalized nanosheets were added to the matrix of PU for enhancing its flame retardant and mechanical property, as contributed from the layered structure and high P content and mechanical robustness of the nanosheets. As observed in TGA analysis, the fraction of ionic liquid on the surface of the nanosheets was only 6.2 wt%. However, the crystal structure of BP nanosheets was found to be stable even after functionalization with the ionic liquid, as confirmed by the XRD peaks appearing at 16.6, 27.3, 34.4, 35.1, and 52.3° (2θ) corresponding to (0 2 0), (0 2 1), (0 4 0), (1 1 1), and (0 6 0) planes of orthorhombic BP crystal. In the Raman spectra of the self-assemblies of BP nanosheets with ionic liquid, the typical peaks for the nanosheets were shifted to a lower wavenumber, showing a red-shift phenomenon. Obviously, the structural properties correlations in these composite were affected significantly, particularly with regard to their applications as flame retardants. Further analytical results from Raman and digital photos of the char residue (Figure 6a–c) have revealed that the degree of graphitization can be decreased from 0.35 to 0.29 or 0.31 for pure thermoplastic PU and composite, respectively (Figure 6d–f) [107,108].

The degree of graphitization was estimated from the intensity ratio of the peaks for disorder and order structures in the hexagonal graphitic layers, corresponding to the D band (~1360 cm^−1^) and G band (~1605 cm^−1^) in the Raman spectra, respectively [107].

In another example, the hybrids or composites of BP nanosheets with polydopamine and polyvinyl alcohol were developed for enhancing the flame retardant and mechanical properties of polyvinyl alcohol. In this strategy, shown in Figure 7, polydopamine or PDA, a bio-polymer with a superior structural stability and adhesive properties, was adsorbed on the surface of BP nanosheets under weakly alkaline conditions by using its building block protonated dopamine to produce PDA-encapsulated BP (BP-PDA) [111].

BP-PDA was added to the matrix of polyvinyl alcohol (PVA) to evaluate the flame-retardant and mechanical properties of its films. The microscale combustion calorimeter (MCC) of PVA/BP-PDA nanocomposite films has demonstrated a reduction in the heat release rate (HRR) and total heat release (THR), as the BP-PDA loading increased from 0.5 to 5 wt% (Figure 8) [111].

It was further concluded from the above results that the BP-PDA extends fire stability to PVA as compared to pure BP. Due to the interfacial interactions between PDA and PVA, the fire-retardant capacity of the nanocomposites increased significantly [112,113,114].

In Table 3, the properties of graphene and polymers-based fire retardants are depicted in order to compare their performance with BP polymers-based fire retardants. For BP/IL/TPU composites, the higher heat release rate and residual char fraction indicated their better performance.

### 3.2. Hybrids of Polymers and BP Nanosheets as Supercapacitors

Although the 2D BP nanosheets are already being used as the material of choice for supercapacitors and other energy storage devices due to their light weight, outstanding power density, superior long-term stability, and safety [116,117,118]. However, in order to modulate and improve their energy densities over a longer period of time, conducting polymers, such as semiconducting ones, with redox-active properties have been customized together with BP nanosheets for producing a hybrid nanomaterial with a high supercapacitive performance. As per one such example, the BP nanosheets or phosphorene were functionalized with polypyrrole by following the ball mill method to attach urea molecules in the first step and the polymerization of carboxylated pyrrole onto FP nanosheets using a FeCl_3_ initiator in a successive step (Figure 9) [119]. The investigations of their supercapacitive properties by electrochemical impedance spectroscopy (EIS) have indicated that the functionalized BP nanosheets with polypyrrole acted almost as an ideal capacitor, as evidenced by the vertical slope in the low-frequency region (Figure 10). The cyclic voltammetry (CV) curves and specific capacitances of the FPPY hybrid nanomaterials have also suggested an enhancement in the specific capacitance to 176.0 F g^−1^ when the PY/FP ratio was changed (Figure 10a).

These functionalized BP nanosheets with polypyrrole have shown a specific capacitance four times higher (411.5 F g^−1^) as compared to that of the pristine PPY, whose capacitance was 106 F g^−1^ (Figure 10b).

The capacitance retention or cycle stability has also increased by a factor of >2 (56.5%), demonstrating their high performance as supercapacitors (Figure 10d) [114]. In a separate development with regard to the supercapacitors, the authors have claimed to obtain a free-standing film of BP nanosheets and polypyrrole (Figure 11). In this unique method, a free-standing film was prepared via a facile one-step electrochemical polymerization method [120]. Herein, the as-prepared BP nanosheets were allowed to be captured by the polypyrrole chains produced during the electrochemical polymerization of pyrrole monomer dispersed into an electrolyte (Figure 11a). These hybrids were then deposited onto a surface of ITO (indium tin oxide) to fabricate a film which, in a subsequent step, was peeled off after a continued electrochemical polymerization process. A large area (5 × 5 cm^2^) film of the PPy/BP film was found to be robust with an excellent mechanical strength and flexibility [120].

The electrochemical properties of a free-standing film studied by CV have revealed a low resistance and good capacitive behavior, with an excellent reversibility in the charge/discharge process (Figure 12). Moreover, the free-standing super capacitor films also retained their flexibility under different bending angles (0°–180°) even after fabrication in a device (Figure 12c). Further observations from the CV curves of the device indicated no obvious degradation, and the corresponding capacitance was almost unchanged, exhibiting a superior mechanical flexibility and excellent structural integrity.

After 10,000 charging/discharging cycles (Figure 12e), the cycling stability of this PPy/BP-based SC was found to be without any decay, pointing towards the compact configuration of the film (Figure 12d), showing exceptional gravimetric (based on the total mass of active material) and volumetric (combined volume) capacitance, achieving high values of 452.8 F g^–1^ and 7.7 F cm^–3^ at a current density of 0.5 A g^–1^, respectively [119]. In Table 4, the properties of supercapacitors made up of the nanocomposites of BP/polymers and graphene/polymers are shown in order to compare the efficiency of two different systems.

The capacitance for a free-standing film of BP nanosheets and polypyrrole was found to be 452.8 F g^–1^, whereas for graphene and polypyrrole it was 417 F g^−1^, indicating the superior efficiency.

### 3.3. Hybrids of Polymers and BP Nanosheets as Ionic Batteries

BP nanosheets have attracted further attention for their usage in Li and Na ionic batteries due to their high theoretical capacity and low working potential [123,124,125]. The sufficient interlayer space in BP nanosheets allows a feasible intercalation and diffusion of metal ions, prompting the scientific community to exploit BP nanosheets for green energy production using ionic batteries. In this regard, a new type of Li-ionic conducting composites of BP nanosheets have been proposed in which a ternary polymer electrolyte containing poly(ethylene oxide) (PEO)/glycol dimethyl ether (TEGDME)/1-ethyl-2,3-dimethylimidazolium bis(trifluoromethylsulfonyl)imide (EMIM-TFSI) and Li-TFSI as lithium salt was combined with the passivated BP nanosheets to establish a correlation between the composition of BP nanosheets and ionic conductivity (Figure 13) [126].

The structural integrity and stability of BP nanosheets were maintained after passivation using the above method. BP nanosheets of over ≈200 nm in lateral size and thickness of 1–8 nm were formed, as evidenced by the TEM images (Figure 13a). The residual Au/Sn alloy and Si contamination were eliminated after exfoliation (Figure 13b,c). The binding energies in the XPS spectra at 529.7 and 529.4 eV were associated with the signals for O 1 s in BP crystal and passivated BP respectively (Figure 13c). The composition distribution of elements such as C, F, O, S, and P was done by EDS mapping. The polymer backbone was mainly responsible for C atoms, whereas the uniform distribution of the 1-ethyl-3-methylimidazolium bis(trifluoromethylsulfonyl)imide (EMIM-TFSI) and Li-TFSI salts was originally pointed towards the fluorine (F) and sulfur (S) [125]. The uniform distribution of phosphorus and oxygen linked to the passivated BP nanosheets additives and electrolyte components, as shown in the density map, explained that the synthesis of the BP nanocomposite electrolyte was homogenous [126].

The temperature-dependent ionic conductivity and other electrochemical properties, such as the electrochemical stability window (ESW) and electrical conductivity (EC), were evaluated for above the BP nanocomposite electrolyte (Figure 14). The highest ionic conductivity, about 2.4 × 10^−3^ S cm^−1^ for CPE-0.5P, was observed and related with the facilitated intercalation and diffusion of Li^+^ in BP nanosheets and the polymer network (Figure 14a) [121]. A current change at 5.5 V (vs. Li/Li^+^) suggested their oxidative electrochemical stability in a wide range of voltage. To prove whether the conductivity is mainly originated from the Li^+^ upon the addition of BP nanosheets, the corresponding EC plots with a polarization voltage of 1 V have shown a considerably low electrical conductivity (EC) (Figure 14b). From these tests, it was confirmed that the ≈10^6^ times lower electrical conductivity of these composites assisted ionic transport in a safe network without any short circuit [126].

### 3.4. Hybrids of Polymers and BP Nanosheets in Optoelectronics

The semiconducting properties of BP nanosheets in the combination of suitable (non)conducting polymers have been also exploited for applications in optoelectronics, such as random-access memories, light-emitting diodes (LEDs), and laser technology [127,128,129,130,131,132]. Obviously, for further improving the combined performance of such systems, the BP nanosheets must be decorated or linked with the well-designed conducting polymer chains that can not only protect the BP nanosheets from the oxidative degradation, but also do not allow any compromise with the properties of either the BP nanosheets and their own. The few reports describing such optoelectronic active nanocomposites of BP/conducting polymers, along with their performance, are highlighted here. A study focusing on flexible photodetectors has proposed a novel method for modifying the BP nanosheets with diluted polymer ionic liquids.

**Figure 13 polymers-15-00947-f013:**
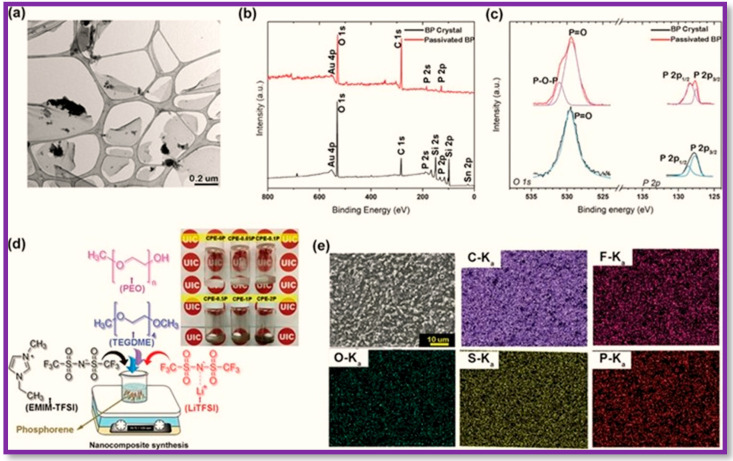
Preparation of the nanocomposite polymer electrolyte with passivated BP nanosheets additive. (**a**) TEM micrograph of the BP nanosheets, (**b**) survey XPS spectra, and (**c**) high-resolution XPS spectra of the P 2p and O 1 s signals of the pristine BP crystal and passivated BP nanosheets. (**d**) Overall synthesis procedure. Inset: photograph of the developed electrolytes. From left to right: CPE-0P (no additive) and nanocomposite polymer electrolytes with 0.05, 0.1, 0.5, 1, and 2 wt% of passivated BP nanosheets. (**e**) SEM image of the 0.5 wt% with the corresponding EDS mapping. Copyright 2020 WILEY-VCH Verlag GmbH & Co. KGaA, Weinheim, Reproduced with permission from Adv. Funct. Mater. [128], Copyright 2020 WILEY-VCH Verlag GmbH & Co. KGaA, Weinheim, Germany.

**Figure 14 polymers-15-00947-f014:**
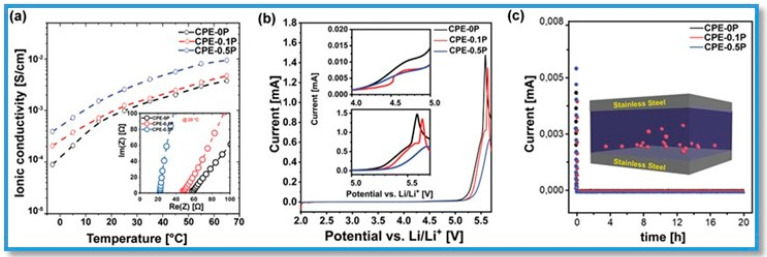
Electrochemical evaluation of the developed electrolytes. (**a**) Ionic conductivity as a function of temperature for CPE–0P, CPE–0.1P, and CPE–0.5P. The inset graph shows the Nyquist plots corresponding samples at 25 °C. (**b**) Linear sweep voltammetry showing electrochemical stability window. (**c**) Direct current polarization tests to measure the electronic conductivity of the developed electrolytes. Copyright 2020 WILEY-VCH Verlag GmbH & Co. KGaA, Weinheim, Reproduced with permission from Adv. Funct. Mater. [128], Copyright 2020 WILEY-VCH Verlag GmbH & Co. KGaA, Weinheim, Germany.

In this noncovalent functionalization method, highly charged polymer ionic liquids (PIL), such as P ([VPIm]Br), P ([VPIm]PF_6_), and P ([VPIm]TFSI), were used to passivate the BP nanosheets in the liquid phase (Figure 15) [128]. The unique feature of this approach included the exfoliation of BP nanosheets in the presence of diluted polymer ionic liquids instead of ionic liquid alone. Here, using the polymer ionic liquids for exfoliation has shown a clear advantage over the stability and aggregation of BP nanosheets.

A comparative study on the effect of the exfoliation and absorption intensity of the suspension of BP nanosheets in diluted polymer ionic liquids and two imidazole small molecule ILs, [EmIm]BF_4_ and [BmIm]BF_4_, has suggested that the exfoliation of BP was improved significantly (Figure 16). Depending on the high intensity and the darkest brown color of the suspension, it was concluded that the BP/P ([VPIm]TFSI) and DMF produced a higher yield of BP nanosheets wrapped with the relevant polymer ionic liquid (Figure 16b). From the Raman spectra and XPS binding energies data, it was clear that the BP nanosheets were protected very well with the polymer ionic liquid chains, thereby excluding any oxidative degradation. For fabricating the photodetector devices, a concentrated solution (≈0.64 mg mL^−1^) of BP/P ([VPIm]TFSI) nanosheets was deposited on the surface of indium tin oxides/polyethylene-terephthalate (ITO/PET) substrate by the drop casting method in order to obtain a thin film with the thickness of ≈5 µm, a width of ≈0.9 cm, and a length of ≈0.6 cm. The photocurrent response of the device under the bias voltages and with the light source power intensity of around 20 mW cm^−2^ was recorded [127]. An increment in the photocurrent density from ≈5 nA cm^−2^ under 0 V bias to 51 nA cm^−2^ under 3 V was observed, highlighting the presence of electron–hole in the BP/P ([VPIm]TFSI) composites (Figure 17a). The consistency in the responsibility of the BP/P ([VPIm]TFSI)-based flexible photodetector device was found to be 4.6 µA W^−1^, indicating no severe effect on the electrical performance of BP nanoflakes, and offering these materials as potential candidates for flexible optoelectronics [131].

Furthermore, when an external bending stress was applied to the device, its photocurrents with the corresponding bending curvatures did not change under dark and light condition, proving its outstanding electrical stability (Figure 17b). The further photostability of the device was supported from the investigation of the time-dependent photocurrent density, which was improved up to ≈50 nA cm^−2^ when exposed to an ambient environment for 120 h (Figure 17d) [131].

In another work, the authors have reported using poly(2-hydroxyethyl methacrylate)-*co*-poly(styrene) (PHMA-*co*-PS) for passivating and encapsulating BP nanosheets and achieving greatly improved electrical transport properties at an electric high field for the BP-based transistor (Figure 18) [132].

Here, a block copolymer based on the poly(2-hydroxyethyl methacrylate) (PHMA) and styrene monomer was chosen for a passivation purpose, owing to its good thermal stability and hydrophobicity (Figure 18a) [132]. Well-defined copolymer PHMA-*co*-PS was synthesized by following reversible addition fragmentation chain transfer polymerization. The resulting polymer with a low polydispersity (PDI = 1.26) possessed a high contact angle (θ = 75°) and thermal stability up to 347 °C (Figure 18c). As shown in the Scheme above (Figure 18), the polymer was used to encapsulate the layers of BP nanosheets and Al_2_O_3_ that were previously implemented on the surface of the atomic layer deposited (ALD), with HfSiO acting as the gate dielectric (Figure 18). The channel length of the gate dielectric was 0.16 μm, as estimated from the scanning electron microscopy image of a BP device. The electrical properties of the BP nanosheets in the FETs device were also evaluated before and after their encapsulation with the PHMA-*co*-PS layer. The *I*_d_–*V*_g_ curves of BP FETs with three different thickness at *V*_d_ = −0.05 V for the 0.16 μm channel length have shown an improvement in on/off current ratios for the corresponding thickness (Figure 19a). Further experiments have suggested that the current on/off ratio increased by a factor of 4–10 after the BP nanosheets were coated with the polymer (Figure 19b). The temperature-dependent on/off current ratio of the device with PHMA-*co*-PS encapsulation was also found to be larger compared to the bare BP nanosheets [132].

The threshold voltage (*V*_th_) of BP FETs as a function of temperature has shown no deviation and remained almost 0.8–1.2 V. after encapsulation, while for bare BP nanosheets, the *V*_th_ decreased, providing insights regarding the charged impurities such as H_2_O and O_2_ on the BP surface that were eliminated by the passivation; hence, the *V*_th_ was stabilized (Figure 19c,d). Clearly, BP nanosheets encapsulated with the polymer layer have demonstrated a better stability with almost unchanged electrical properties that can be harnessed in FET device applications in ambient conditions [132].

### 3.5. Hybrids of Polymers and BP Nanosheets as Drug Delivery Platform

As compared to other 2D nanomaterials such as graphene, transition metal dichalcogenides (molybdenum disulfide), and Mxene, the phosphorene or BP nanosheets can offer a superior biocompatibility and a great biodegradability in the physiological environment owing to its reactivity with the water and oxygen; therefore, its applications can be extended to the biomedical field e.g., phototherapy, drug delivery, biosensing, theranostics, and bone regeneration [133,134,135,136,137,138]. Some of the examples describing the strategies and applications of BP nanosheets with conjugated polymers in biomedical sciences are summarized here. In one such case, the surface of the BP nanosheets were fabricated with the poly-L-lysine (PLL) or poly(ethylene glycol) (PEG) linker together with the two different fragments of peptides, namely RGD and KRK (Figure 20), for using the assembly against breast cancer [139].

In this two-step process, first, the covalent bioconjugation of the peptide with the poly-L-lysine (PLL) or poly(ethylene glycol) (PEG) was carried out, followed by the non-covalent immobilization of PLL-peptide or PEG-peptide to the surface of BP nanosheets (Figure 20). Finally, the effect of peptides RGD and KRK in combination with functionalized BP nanosheets as potential drugs in breast cancer therapy was investigated. The analysis of functionalized BP nanosheets with both the polymers was done by means of Raman and SEM to examine the morphology [139].

As shown in the Figure 21A, the several vibrational bands in the range of 3600–3300 cm^−1^, 3100–2200 cm^−1^, 1750–1550 cm^−1^, 1550–1300 cm^−1^, and 1300–900 cm^−1^ appeared for the FLBP-PEG-peptide assembly. A characteristic narrow and intense band at 1112 cm^−1^ was associated with the C–O–C group in FLBP-PEG, which was not observed for FLBP-PLL-peptide conjugates (Figure 21A). The presence of amino groups was confirmed with the presence of a corresponding band at 1300–900 cm^−1^. The further morphological analysis before/after functionalization under Scanning electron microscopy has revealed that the non-ionic linker PEG assisted in the interconnection of adjacent layers, whereas the positively charged PLL induced the formation of dispersion throughout the sample (Figure 21(Bb–e)). Nevertheless, the above tests proved the formation of bioconjugates successfully. To target the breast, e.g., breast cancer cell lines, MCF-7, MDA-MB-231, and HB2 cells, these conjugates of BP nanosheets and PEG/PLL were scrutinized for their cytotoxicity (Figure 22).

The treatment of breast cancerous cells with the different concentrations of these bioconjugates has indicated a loss in viability among the MCF-7 and MDA-MB-231 infected cells at 4 μg ml^−1^ [135]. Before functionalization, BP nanosheets exhibited toxicity for HB2 (normal mammary cells) at a concentration of (20 μg mL^−1^) (Figure 22a). However, after modification, BP nanosheets were found to be less harmful towards the normal mammary cells; HB2 with ~70% cell viability at 20 μg mL^−1^ (Figure 22b). The modified BP nanosheets are more toxic to the triple-negative breast cancer cell culture MDA-MB-231 (viability < 34%) compared to the luminal breast cancer cell culture MCF7 (viability < 46%). From these results, it can be concluded that the decoration of the BP nanosheets surface definitely extends their biocompatibility with regard to their biomedical applications and the possibilities of using these bioconjugates as drug delivery system against cancer [139].

### 3.6. Hybrids of Polymers and BP Nanosheets for Tissue Engineering

Following up with the biomedical usage of BP nanosheets, a research group has combined polymers-based hydrogels with BP nanosheets to improve the mineralization of CaP crystal nanoparticles for bone-tissue regeneration. In this combinatorial approach, double-network (DN) hydrogels based on three distinct polymers were generated via photo polymerization (Figure 23) [140].

As shown in the Figure 23(Aa), the biocompatible synthetic polymers such as [poly(2-hydroxyethylacrylate) (PHEA), poly(*N*,*N*-dimethyl acrylamide) (PDMA), or polyacrylamide (PAM)], were cross-linked in combination of one of three methacrylate-decorated natural polymers to form the nanoengineered network with a high mechanical strength and tunable toughness. In these networks, BP nanosheets were introduced, adding multiple functions to the exceptional mineralized matrix’s formation ability and excellent bioactivity to the hydrogels. Covalent bonds alone or together with hydrogen bonds in a network weaken the strength of DN hydrogels, whereas the presence of covalent bonds and the high density of polymer chain entanglements offered a high strength to DN hydrogels (Figure 23(Ab)). The successive addition of BP nanosheets into the DN hydrogels resulted in NE hydrogels that induced the formation of the CaP matrix for bone regenerations (Figure 23(Ab)). The networks with outstanding mechanical properties were able to withstand high levels of compression in longitudinal and transverse directions even at an extremely high strain. However, upon the removal of the compression force, the initial cylindrical shapes were regained by both the DN and NE hydrogels. These tough DN and NE hydrogels were also resistant towards and further deformations induced by bending (Figure 23(Bb), upper right), knotting (Figure 23(Bc), upper right), and crossover/knotting stretching (Figure 23(Bd)) without any rupture, indicating a high degree of strength, stiffness, and toughness.

The ability of these DN gels to regenerate the bone was evaluated in vivo in the rat calvarial defect model. Under micro-computed tomography (micro-CT) imaging and from the quantification of the bone volume/tissue volume ratio (BV/TV) (Figure 23C(a,b)), including the studies of bone mineral density (Figure 23(Cc)), it was suggested that these BP nanosheet-encapsulated NE hydrogels (PAM/ChiMA/BP and PAM/AlgMA/BP) significantly improved bone regeneration. As the mineralization occurred within NE hydrogels, the amount of newly formed bone also increased simultaneously [140,141].

### 3.7. Hybrids of Polymers and BP Nanosheets for Bioimaging and Photothermal Cancer Therapy

It is very well known that cancer is responsible for millions of deaths per year across the globe. Until today, considerable efforts have been ongoing to develop an effective therapy against this terminal disease. Currently, two unique therapies, namely photothermal therapy (PTT) and photodynamic therapy (PDT), are gaining a tremendous amount of attention as they can offer a good efficiency without any major injury. In PTT, an active material system absorbs near-infrared (NIR) light and turns it into heat under light irradiation, leading to the elimination of cancer cells, whereas in PDT, the reactive oxygen species (^1^O_2_) are generated during the photosensitizing process in order to kill the infected cells. In PDT, the efficiency of the system depends upon the concentration of ^1^O_2_ species which are reduced under a hypoxia environment in tumors. Therefore, by combining the PTT with PDT, this issue can be resolved as the heat in the PTT process can enhance the blood flow and thereby increase the oxygen supply for an increased ^1^O_2_ production. One such successful attempt to combine PTT and PDT is summarized here, in which BPQDs (black phosphorus quantum dots) smaller than 10 nm were functionalized with water-soluble PEG for enhancing their biocompatibility, excellent NIR photothermal performance, and ^1^O_2_ generation capability (Figure 24) [141].

First, the BPQDs with a smaller size were obtained by liquid exfoliation methods (Figure 24a). These as-obtained BPQDs were decorated with the water-soluble PEG-NH2 via electrostatic bonding. To this, a photosensitizer known as rhodamine B (RdB) was conjugated to obtain the RdB/PEG-BPQDs that were analyzed by UV–Vis and Raman spectrometry to confirm the functionalization (Figure 24b,c). These RdB/PEG-BPQDs were evaluated for their anti-cancerous capabilities in 4T1 tumor-bearing Balb/c mice under in vivo conditions [141].

The real-time temperature changes in tumors were monitored under the infrared thermal camera (Figure 25a). The tumors injected with the PEGylated BPQDs (groups V and VI) have shown a considerable increment in temperature within 2 min (Δ*T* ≈ 30 °C) under NIR irradiation, as compared to the group III that was treated only with a laser. These observations indicated towards the excellent photothermal efficiency of PEGylated BPQDs [141].

Other studies involving the biodegradable BPQDs functionalized with poly (lactic-co-glycolic acid) reported an enhanced photothermal stability for cancer therapy (Figure 26) [142].

In this study, BPQDs as small as 3 nm were loaded into poly (lactic-co-glycolic acid) (PLGA) and processed by an oil-in-water emulsion solvent evaporation method to produce ∼100 nm BPQDs/PLGA nanospheres (NSs). Owing to the biodegradability and biocompatibility of PLGA, encapsulated BPQDs increased the blood circulation and effective accumulation in tumors for an effective cancer therapy. The biodegradability tests of control BPQDs and BPQDs/PLGA NSs in the phosphate-buffered saline (PBS; pH 7.4) have revealed that after functionalization, BPQDs retained their stability and performance during the initial 24 h (Figure 26a,b) in comparison to bare BPQDs. After 8 weeks, the absorption and photothermal behavior of the BPQDs/PLGA NSs started to diminish because of the degradation of PLGA under similar conditions. According to the measurements done for residual weight over the time period, the BPQDs/PLGA NSs displayed a negative trend during first week; however, further rapid weight loss occurred after 8 weeks (Figure 26c). The degradation behavior was also observed visually under SEM and TEM. During the first week, the NSs retained their shape and morphology, however, after 8 weeks, they appeared to degrade, resulting in a total disruption in their morphology (Figure 26d, SEM and TEM images). An illustration of the degradation is shown in Figure 26, in which the PLGA shell degrades under acidic conditions via desertification, leading to the evolution of water and carbon dioxide. Such degradation induces the disintegration of NSs, exposing the BPQDs to air and water directly, which further results in the formation of byproducts such as nontoxic phosphate and phosphonate [142].

The in vivo cytotoxicity investigations of these BPQDs/PLGA NSs were also conducted first on the healthy female Balb/c mice. Compared to the control group, the standard hematology markers, e.g., white blood cells, red blood cells, hemoglobin, mean corpuscular volume, mean corpuscular hemoglobin, mean corpuscular hemoglobin concentration, platelets, and hematocrit, were found to be normal, with no major difference pointing towards the absence of an obvious infection and inflammation in the treated mice (Figure 27a). For the blood analyses, the parameters, including alanine transaminase, aspartate transaminase, total protein, globulin, total bilirubin, blood urea nitrogen, creatinine, and albumin, appeared to be normal as well (Figure 27b). These observations have indicated that NSs treatment does not affect the chemical nature of blood in mice [136]. The histological changes in the tissue obtained from the liver, spleen, kidney, heart, and lung were also investigated. The corresponding tissue stained with eosin have shown no signal for any organ damage during the period of treatment (Figure 27c). These results have suggested that BPQDs/PLGA NSs are biocompatible and have no toxicity towards healthy cells [142].

In Table 5, the therapeutic properties of the nanocomposites of BP polymers and graphene polymers obtained for a different test or model with their relevant mode are summarized. The nanocomposites of PEGylated BPQDs and PLGA–BPQDs were evaluated against the breast cancerous cells and were found to be effective and highly efficient during photothermal and photodynamic therapy.

### 3.8. Hybrids of Polymers and BP Nanosheets for Bacteria Capture and Elimination

Pollutants or contaminants pollute drinking water; due to which, some parts of the world are facing drinking water scarcity. The byproducts produced during the disinfection process via the currently used purification methods can be environmental hazards. In a very recent report, a group has made efforts to design a rather environmentally friendly disinfection method by using a thermo-responsive poly(N–isopropyl acrylamide)–functionalized black phosphorus (named BP–PNIPAM) to capture and eliminate bacterial cells under the stimuli of near-infrared irradiation. An overview of this thermoresponsive composite, followed by its capacity for capturing *E. coli* (A and B) and S. aureus, is shown in Figure 28 [145]. Here, the thermo-responsive PNIPAM brushes were linked on to the surface of BP nanosheets by following in situ atom transfer radical polymerization (ATRP). Upon irradiation, these brushes can undergo a hydrophilic-to-hydrophobic transition in order to capture the bacterial cells in water (Figure 28a).

The ability of these nanocomposites to capture the bacterial strains such as *E. coli* and *S. aureus* was examined via the colony counting method. A concentration of BP–PNIPAM, for instance higher than 50 μg mL^−1^, was found to be sufficient for capturing the 80% colony of E. coli. However, for S. aureus, the capturing efficiency was only 25.16%. The bacterial capture efficiency of BP–PNIPAM composites further increased to 94.25% at 200 μg mL^−1^ (Figure 28b,d).

As per the proposed mechanism for the bacterial capture, in first step, the hydrophobic PNIPAM binds to the surface of bacterial cells via non-specific interactions, and in turn, the bacterial cells become wrapped into the BP–PNIPAM aggregation as triggered by the irradiation. Scanning electron microscope images have demonstrated that the BP–PNIPAM aggregations were able to capture both Gram-negative *E. coli* and Gram-positive *S. aureus* successfully (Figure 28c,e). Further tests under near-infrared (NIR) irradiation have indicated that these BP–PNIPAM aggregations could eradicate both *E. coli* and *S. aureus* for the BP–PNIPAM concentrations of 100 and 200 μg mL^−1^ and thus could be a promising candidate for a safe application in water disinfection [145].

## 4. Conclusions

Since the elusive properties of black phosphorus or phosphorene came to light, the researchers working on 2D materials have shifted their focus to exploring this fascinating material, its modulation, and optimizing its properties. There are few methods in which the different shape and size of phosphorene can be obtained with the precise control over its electronic characteristic, as explained in the article. However, for the processing and fabrication of different nanostructures of phosphorene, such as nanosheets or nanoribbons, it has to be coupled with a polymer in order to prevent its degradation without compromising its features for the applications. Various attempts to combine the phosphorene nanosheets or quantum dots with a suitable and active polymer chain have been carried out successfully. Active polymers (ionic) conducting and non-conducting, such as poly(vinyl alcohol), poly(ethylene glycol), poly(imidazole), poly(styrene sulfonic acid), poly(pyrrole), poly(dopamine), and polypeptides, have been linked to the surface of phosphorene nanosheets via a covalent linkage or adsorption process. In these composites, the oxidation or degradation of nanosheets was prevented and the solubility of phosphorene in common organic solvents was increased for a better processability. Some of these hybrid materials or composites appeared as hydrogels and have provided a superior biocompatibility to the phosphorene nanosheets. After their thorough chemical analysis, these composites have been fabricated and employed in a wide range of applications that include fire retardants, capacitors, ionic batteries, field effect transistors, drug delivery, and cancer therapy. The efficiency of capacitors and batteries was found to be higher. For cancer therapy, the cell toxicity experiments have proven that some of the biocompatible composites of phosphorene nanosheets and a relevant polymer can effectively be used for treating the tumor in mice. Some of the BP polymers nanocomposites summarized here can also be applied for the water disinfection process, in which the bacterial cells are captured and eradicated under irradiation. Obviously, the progress made so far with regard to the BP polymers nanocomposites has caused a greater impact, thereby reducing the burden on other similar 2D nanomaterials, such as graphene, for multiple applications. The future of BP polymer nanocomposites appears to be bright; we will witness their usage in space technology as well.

However, despite significant progress with the composites of phosphorene and polymers, there is still an improvement with regard to their robustness and performance. The high-scale production of phosphorene nanoobjects with a controlled shape and structure is highly demanded. Second, multiple polymers, with a superior performance and the ability to synergize with BP nanoobjects, are to be developed. Last but not the least, the fabrication of these hybrid composites for a commercialization purpose or for real-life applications would need technological advancements which, as of yet, have not been developed.

## Data Availability

Not applicable.

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
