# Peer review of "Towards the Future of Polymeric Hybrids of Two-Dimensional Black Phosphorus or Phosphorene: From Energy to Biological Applications"

_polymers, 2023, doi:10.3390/polym15040947_

Round 1
Reviewer 1 Report
This review article is a collection of low quality copies of figures from other articles they cited. I do not see any value of this review to this area of research which is still evolving. This article looks like summarizing other groups' work not a review of the field.
Reviewer 2 Report
In this manuscript, the authors reviewed a new 2D material, black phosphorus (BP), with properties such as, higher carrier mobility, biocompatibility, thickness dependent band gap, and optoelectronic characteristics that can be harnessed for multiple applications. With the integration with other polymers, the robustness and the performance can be improved, which can meet the on-going optoelectronics needs for commercialization purposes. The review is very interesting and the content is very well organized. Considering the possible application fields the topic is rather timely and deserves to be considered for publication after minor revisions.
Suggested revisions:
1. In the Abstract, the authors emphasized that the link with the suitable and functional organic counter macromolecule can provide a protection from air/water of BP, so the authors can also provide more examples in the main manuscript, such as after line 298 or in the other suitable paragraphs.
2. There are many typos in units, such as line 33 (cm(2)/V·s), line 83 (200 C), line 125 and line 127 (cm−1), line 229 (to he D), line 269 (F g), line 278 (F g−1), line 432 (μA W−1 u), and also many similar typos in the figure legends. There are other typos, such as line 23 (ofering), line 469 (Al2O3).
3. The contents from line 593 to line 595 should be in somewhere of the figure legends.
4. Some abbreviations for the same words have been shown repeatedly in the main manuscript, such as 3 times of near-infrared (NIR), 2 times of poly(ethylene glycol) (PEG), 2 times of bone mineral density (BMD), and so on. Also, there are 2 times of scanning electron microscopy (SEM), but there is no abbreviation for TEM.
Reviewer 3 Report
I would like to recommend its publication upon a minor revision. Some comments could be found as follows.
1. Since the exploration of the hybrid system of BP nanosheets and active polymers aims at replacing graphene in multiple applications, it is recommended to provide tables with property data for illustrating the comparison between graphene and the developed composites/hybrid system.
2. It is recommended to provide a table showing the comparisons of different methods to synthesize the BP nanosheets or phosphorene.
3. It is recommended to provide some principles for selecting active materials for the BP nanosheet‒based hybrid system. Are there any requirements for suitable active polymers? What kind of active materials can maximize their synergistic effect on the properties?
4. For each type of application, it is recommended to summarize the reported materials and their properties in a table.
5. A comprehensive but precise discussion on the flame-retardant properties of the BP hybrid system could be provided by discussing the flame-retardant mechanism. Some related works could be helpful. For example, Hou, Y., Xu, Z., An, R., Zheng, H., Hu, W., & Zhou, K. (2022). Recent progress in black phosphorus nanosheets for improving the fire safety of polymer nanocomposites. Composites Part B: Engineering, 110404.
6. Some typos should be corrected.
Reviewer 4 Report
Dear Author,
I studied your manuscript entitled "Towards the future of polymeric hybrids of two-dimensional black phosphorus or phosphorene: From energy to biological applications". This paper comprises interesting results that certainly deserve publication. I recommend it for publication in the Journal of Polymers as is.
Round 2
Reviewer 1 Report
The authors included my comments and one of the reviewers who provided suggestions to improve the paper's quality.